# A Novel Classification of Endometriosis Based on Clusters of Comorbidities

**DOI:** 10.3390/biomedicines11092448

**Published:** 2023-09-02

**Authors:** Antonio Sarria-Santamera, Yerden Yemenkhan, Milan Terzic, Miguel A. Ortega, Angel Asunsolo del Barco

**Affiliations:** 1Department of Biomedical Sciences, Nazarbayev University School of Medicine, Astana 010000, Kazakhstan; 2Department of Medicine, Nazarbayev University School of Medicine, Astana 010000, Kazakhstan; yerden.yemenkhan@nu.edu.kz; 3Department of Surgery, Nazarbayev University School of Medicine, Astana 010000, Kazakhstan; milan.terzic@nu.edu.kz; 4Clinical Academic Department of Women’s Health, National Research Center for Maternal and Child Health, University Medical Center, Astana 010000, Kazakhstan; 5Department of Obstetrics, Gynecology and Reproductive Sciences, School of Medicine, University of Pittsburgh, Pittsburgh, PA 15213, USA; 6Department of Medicine and Medical Specialties, Faculty of Medicine and Health Sciences, University of Alcalá, 28801 Alcalá de Henares, Spain; miguel.angel.ortega92@gmail.com; 7Ramón y Cajal Institute of Sanitary Research (IRYCIS), 28034 Madrid, Spain; 8Cancer Registry and Pathology Department, Hospital Universitario Principe de Asturias, 28805 Alcalá de Henares, Spain; 9Department of Surgery, Medical and Social Sciences, Faculty of Medicine and Health Sciences, University of Alcala, 28801 Alcalá de Henares, Spain; 10Department of Epidemiology and Biostatistics, Graduate School of Public Health and Health Policy, The City University of New York, New York, NY 10017, USA

**Keywords:** adults, endometriosis, cluster analysis, female

## Abstract

Endometriosis is a heterogeneous, complex, and still challenging disease, due to its epidemiological, etiological and pathogenic, diagnostic, therapeutic, and prognosis characteristics. The classification of endometriosis is contentious, and existing therapies show significant variability in their effectiveness. This study aims to capture and describe clusters of women with endometriosis based on their comorbidity. With data extracted from electronic records of primary care, this study performs a hierarchical clustering with the Ward method of women with endometriosis with a subsequent analysis of the distribution of comorbidities. Data were available for 4055 women with endometriosis, and six clusters of women were identified: cluster 1 (less comorbidity), cluster 2 (anxiety and musculoskeletal disorders), cluster 3 (type 1 allergy or immediate hypersensitivity); cluster 4 (multiple morbidities); cluster 5 (anemia and infertility); and cluster 6 (headache and migraine). Clustering aggregates similar units into similar clusters, partitioning dissimilar objects into other clusters at a progressively finer granularity—in this case, groups of women with similarities in their comorbidities. Clusters may provide a deeper insight into the multidimensionality of endometriosis and may represent diverse “endometriosis trajectories” which may be associated with specific molecular and biochemical mechanisms. Comorbidity-based clusters may be important to the scientific study of endometriosis, contributing to the clarification of its clinical complexity and variability. An awareness of those comorbidities may help elucidate the etiopathogenesis and facilitate the accurate earlier diagnosis and initiation of treatments targeted toward particular subgroups.

## 1. Introduction

Endometriosis is a complex, heterogeneous, and still challenging disease, characterized by the presence of functionally active endometrial-like tissue, stroma, and glands outside the uterine cavity [1]. Despite extensive research, endometriosis is still an enigmatic disease, due to its specific pathogenesis, and epidemiological, diagnostic, therapeutic, and prognosis characteristics [2]. Endometriosis carries a significant economic burden in direct medical care and indirect costs because of its impact on quality of life, social function, and work productivity [3]. Existing therapies show significant variability in their effectiveness, possibly pointing out that treatments should be optimized and personalized [4]. Moreover, there is an increasing recognition that considering endometriosis as a pelvic gynecological disorder does not reflect its true scope and manifestations: endometriosis affects metabolism in liver and adipose tissue, leads to systemic inflammation, and alters gene expression in the brain that causes pain sensitization and mood disorders [5].

Several theories have been proposed to explain the molecular and biochemical mechanisms associated with endometriosis. Retrograde menstruation [6] was the first hypothesis suggested. Genetic predisposition and hormonal factors such as resistance to progesterone, estrogen dependence, inflammation, angiogenesis, vascularization processes, oxidative stress, resistance to apoptosis, and immunological factors would have to be involved in various degrees [7,8,9,10]. Recently, an infectious origin has also been suggested [11]. However, there is no clear explanation of the etiopathogenesis of this condition.

Pain is the main symptom of endometriosis [10]. Usually, it is associated with a menstrual period (dysmenorrhea), but often progresses to include noncyclic pain. Several processes seem to be involved in the pathogenesis of pain, but many aspects are still unclear, as there is no correlation between pain severity and the stage of endometriosis [12,13]. Infertility is also associated with endometriosis [14,15], but the cause–effect connection is still under debate [16].

Overall, the clinical presentation of endometriosis is highly variable and relates poorly to the extent of the disease: endometriosis may present with any degree, character, type, or location of pain, or with no pain at all; at any point in the reproductive age spectrum; with or without preceding symptoms; and may have only non-gynecologic symptoms or a wide range of other symptoms with systemic effects [17]. This variety of symptoms has not been well-characterized yet for all endometriosis patients [18].

The classification of endometriosis is also contentious. Current classification systems are based on the disease’s clinical characteristics, such as anatomy, histology, and prognosis, but are not comprehensive and do not correlate with the diversity and severity of clinical manifestations nor reflect the levels of pain and risk of infertility, and lack prognostic value in predicting the response to treatment or disease progression [19]. 

The diagnosis of endometriosis is also challenging because of the absence of specific biomarkers, and imaging tests have limited sensitivity and specificity [20]. Although the gold standard for diagnosis remains a histological confirmation, the non-surgical diagnosis of endometriosis based on history, physical findings, and imaging (transvaginal ultrasonography; MRI) has been recognized as reliable [21]. The diversity in the clinical course and diagnostic complexities is reflected in the quite-variable estimates of the prevalence of endometriosis [22], ranging from 1 to 5%, while the incidence ranges between 1.4–3.5 per thousand per year, depending on the type of data and the design used for those analyses [23].

A common clinical finding is the association of endometriosis with multiple diseases [24,25], including certain types of cancer, not only ovarian but also endometrial cancers [26]. In-depth knowledge of the comorbidities may help to clarify the clinical complexity and variability of endometriosis, facilitating an accurate earlier diagnosis and initiation of targeted treatments.

Typically, studies investigating comorbidities have explored the association of endometriosis with specific health problems, providing a ‘fragmented’ picture of this complex disease without considering possible inter-relationships between comorbidities. Comorbidities may develop following patterns and, thus, can occur in so-called clusters. Comorbidity cluster analysis is an emerging statistical procedure that has demonstrated value in characterizing chronic diseases [27]. Cluster analysis is a data-driven technique with no a priori theory applied to how we expected the comorbidities to cluster. Individuals sharing the same comorbidity profile or falling into the same comorbidity cluster may share similarities in their disease pathogenesis and clinical characteristics.

This study aims to use data obtained from electronic medical records to identify comorbidity clusters and their frequency in women with endometriosis in Spain and to characterize individuals from different comorbidity clusters depending on age and healthcare utilization variables.

## 2. Materials and Methods

This is a cross-sectional study that will analyze the data extracted from the Primary Care Clinical Database (PCCD) of the Spanish National Health System registered during 2013–2017. Reporting of this study was based on the STROBE statement [28], which aimed to improve and strengthen the dissemination of observational studies. The database analyzed here consists of all visits of women aged between 16 and 65 registered with their primary healthcare centers in 6 Spanish regions. PCCD includes the following information from each patient: age, diagnosis, visits to GP, and referrals to specialists. GP assigns specific diagnoses in the PCCD, which are coded using the International Classification of Primary Care, 2nd edition (ICPC-2) [29]. Diagnostic codes are included in the electronic medical record by GP based on the clinical information they have available from their own care for patients or from reports provided by specialists. ICPC-2 classifies patient data and clinical activity in the domains of General/Family Practice, considering the frequency distribution of problems seen in these domains. It allows classification of the patient’s reason for encounter, the problems/diagnosis managed, interventions, and the ordering of these data in an episode of care structure. ICPC-2 has a biaxial structure and consists of 17 chapters, each divided into 7 components dealing with symptoms and complaints, diagnostic, screening and preventive procedures, medication, treatment and procedures, test results, administrative, referrals, and other reasons for encounters and diseases. For the purpose of this study, we refer to comorbidities as any of the codes referring to any of these 7 components that were registered as diagnosis using the ICPC-2 in women’s health records.

Comorbidities with a frequency higher than 5% were included in the analysis. Clustering was performed by using hierarchical Ward’s linkage cluster analysis. Two cluster analyses were conducted, the first one for identifying groups of women with endometriosis sharing similar comorbidity patterns, the second one for comorbidities that may be grouped because of being commonly diagnosed in the same women. This double approach attempts to identify relatively homogeneous groups of women/cases, as well as homogeneous groups of comorbidities/variables. The algorithm for those classifications starts with each case (or variable) in a separate cluster and combines clusters until only one is left. Distance or similarity measures are generated by the proximities procedure. Subsequently, two among these groups having close similarities were combined and formed as one group, leaving other observations as separate groups. Furthermore, as the hierarchy ascended, the process continued by combining the next two similar groups. With each ascending, heterogeneous clusters were created. Finally, a dendrogram was formed. Based on a visual assessment of the dendrograms obtained, data were finally grouped into specific number clusters.

No ethical review was necessary as PCCP is a publicly available database that includes anonymized information already collected and obtained in routine clinical care. Data are maintained by the Ministry of Health and provided de-identified for research purposes.

## 3. Results

Data were available from 4055 women with endometriosis, aged between 21 and 50. Of these women, 84% had at least one comorbidity. Figure 1 describes the process of obtaining the final data for analysis.

Table 1 shows the main characteristics in terms of age and comorbidities of women with endometriosis (*n* = 4055) included in this analysis. The frequency of comorbidities ranges from 5.01% to 26.14%. Anxiety, headache/migraine, upper respiratory tract infection (URTI), chronic/allergic rhinitis, and contact dermatitis/eczema were the most common comorbidities. Figure 2 illustrates the results of the clustering analysis of 26 comorbidities in the form of a dendrogram using the hierarchical Ward’s linkage method. An imaginary line was drawn at a cut-off value of 11, which resulted in six clusters. We constructed six clusters based on a visual assessment of the dendrogram. Table 2 reflects the frequency of comorbidities and other relevant variables for the six clusters. Figure 3 shows how comorbidities aggregate.

The main characteristics of the six identified clusters of women with endometriosis are the following: Cluster 1 (less comorbidity) has the second largest number of patients (*n* = 1212). It has the lowest GP visits (8.1%). All frequencies of the comorbidities are less than 17% and most comorbidities (*n* = 20) have frequencies between 0 and 4%.Cluster 2 (multiple comorbidities) has the lowest number of patients (*n* = 283) but shows the highest number of visits to the GP (28.3%) and to specialists (20.2%). Most patients are between 41 and 50 (61.48%). Patients in the cluster have a high frequency (>30%), of multiple comorbidities (*n* = 7) (anxiety (73.85%), headache/migraine (68.55%), URTI (51.59%), chronic/allergic rhinitis (48.6%), bursitis/tendinitis (44.88%), anemia (31.1%), and elevated cholesterol (31.8%)).Cluster 3 (anxiety and musculoskeletal disorders) is the largest in terms of patients (*n* = 1334). The cluster has a significantly higher prevalence of anxiety (37.48%), back pain (22.56%), and bursitis/tendinitis (22.26%).Cluster 4 (type 1 allergies or immediate hypersensitivity) has the highest prevalence of chronic/allergic rhinitis (49.06%), asthma (35.58%), and urticaria (30.71%).Cluster 5 (anemia and infertility) has the highest prevalence of anemia (54.68%) and infertility (58.91%).Cluster 6 (headache and migraine) has the highest frequency of patients aged between 21 and 30 (14.96%). The cluster has the highest prevalence of headache/migraine (96.4%).

## 4. Discussion

This work demonstrates the feasibility of using cluster analysis to generate a granular view of the multidimensionality of endometriosis based on comorbidities. Women with endometriosis can be classified into six clusters, according to the presence of comorbidities: cluster 1 (less comorbidity), cluster 2 (multiple comorbidities), cluster 3 (anxiety and musculoskeletal disorders), cluster 4 (type 1 allergy or immediate-type hypersensitivity), cluster 5 (anemia and infertility), and cluster 6 (headache and migraine). Previous studies have suggested links between endometriosis and several comorbid conditions. Findings from this work put forward diverse relationships that may be occurring at clinical, biochemical, or molecular levels. Different comorbidities would be associated with specific biochemical and molecular mechanisms [30,31,32].

Although most women (70%) could be separated into specific clusters according to their comorbidities, there is also a large proportion of women with endometriosis who do not have a significant association with comorbidities (30%), while a small cluster (7%) is composed of multi-morbid patients which also has the highest mean age. Below, we describe the main characteristics of the clusters.

### 4.1. Multiple Comorbidities and Advanced Age Cluster

This cluster includes women of older age as well as a high frequency of abdominal pain and specific comorbidities, including infections (urinary and upper respiratory), metabolic (obesity, hypercholesterolemia, and hypertension), and sleep disorders, but also a high frequency of comorbidities included in the other clusters. This group also shows a higher use of healthcare services, both for GPs and specialists.

### 4.2. Anxiety and Musculoskeletal Disorders Cluster

Anxiety, depression, and a low quality of life are frequently associated with endometriosis and show a high prevalence in women with chronic pelvic pain. This may indicate that psychological factors may influence and alter the degree of the pain sensation in women with endometriosis, and negatively affect the quality of life from chronic pelvic pain. Psychological factors may contribute to the severity of the symptoms and the effectiveness of the treatments, but it is still unclear if psychological comorbidities are a result of endometriosis itself or other factors such as chronic pain [33]; however, it seems that the presence of psychological problems may amplify pain. Genes responsible for pain and anxiety were found to be up-and-downregulated in women with endometriosis, indicating possible brain modulatory effects because of endometriosis [34,35].

Endometriosis-induced sciatica, first described by Schlincke in 1946 [36], is a frequent clinical finding, as well as pain in various areas of the leg, such as the buttock, anterior, and lateral thigh [37]. Endometriosis has also been associated with immune-inflammatory disorders, such as rheumatoid arthritis cursing with intermuscular symptoms, back pain, tendonitis, or bursitis, while musculoskeletal disorders have also been found to be frequently associated with mental problems [38]. The underlying pathophysiological process remains unclear and the explanations provided are mostly at hypothetical levels [30].

### 4.3. Type 1 Allergies/Immediate-Type Hypersensitivity Cluster

A high frequency of family history of allergies, to food, medications, or chemical products, as well as eczema, asthma, and allergic rhinitis, has been found in patients with endometriosis [39,40]. One of the proposed pathophysiological mechanisms could be the presence of pro-inflammatory cytokines (like TNF-α and IL-1β) found in both allergic disorders and endometrial tissues. Macrophages are found abundantly in the peritoneal fluid of patients with endometriosis. These cells are thought to play a significant role in the pathogenesis of endometriosis: TNF-α and IL-1β stimulate ectopic endometrial cells to synthesize histamine-releasing factor (HRF), which is also involved in the pathogenesis of allergy. HRF induces IL-4 and IL-13 production by changing the balance between Th1 and Th2 to a higher Th2 response, which is also a driver of allergic reactions. Many activated mast cells and basophils have been identified in endometrial implants, which may indicate the same pathologic mechanism [41]. However, whether endometriosis triggers the development of allergies or vice versa remains unknown.

### 4.4. Infertility and Anemia Cluster

Approximately 60% of women with endometriosis in this cluster have an association with infertility, but this problem is infrequent in the other clusters. Animal models have demonstrated the relationship between endometriosis and infertility [42], and prospective data from the Nurses’ Health Study II reflected a temporal association between endometriosis and infertility. A diverse list of explanations has been proposed, including pelvic cavity, ovarian, uterine/endometrial, and genetic factors, but the direct causal link for endometriosis infertility has not been clarified [43,44]. Nerve fibers found in the endometrium also present in the myometrium may cause the formation of adhesions in the pelvis and pain. Dysregulated immune activity such as increased complements along with altered gene products could result in improper implantations [45]. Pain may have an influence, in particular, dyspareunia, leading to reduced coital frequency [46], impacting the chances of natural conception [47].

Endometriosis can affect the length and severity of your menstrual cycle and women may experience heavy menstrual bleeding. Heavy menstrual bleeding represents a clinical entity that may result in iron depletion and consequent iron-deficient anemia. In an animal model, endometriosis cases were found to have lower hematocrit, hemoglobin, and mean corpuscular volume [48]. The increased request for iron may be linked with lower fertility [49].

Women with surgically diagnosed endometriosis reported a higher rate of upper respiratory infections. The altered immune function and inflammatory response, commonly present in patients with endometriosis, could be a possible explanation [49,50].

### 4.5. Headache/Migraine Cluster

Women with endometriosis present a high prevalence of migraines; women with migraines present a high prevalence of endometriosis [51]. Diverse hypotheses have been proposed to explain this association. A key role of estrogen has been suggested, as both endometriosis and migraines exhibit a peri-reproductive peak frequency and share estrogen-based risk factors such as early menarche, increased exposure to menstruation, and menorrhagia [52].

Another hypothesis is that the repetitive exposure to prostaglandin and nitric oxide (NO) may cause central and peripheral hypersensitivity. Prostaglandins, one of the triggers that can lead to migraines, have been consistently implied in the pathogenesis of endometriosis [53,54]. Moreover, the production of NO is increased during endometriosis [55]. A possible common genetic predisposition between endometriosis and migraines is another hypothesis [56], which may share possible common inflammatory pathways. The expression of calcitonin gene-related peptide (CGRP) [56], a vasodilator of cerebral vessels involved in the pathophysiology of migraines, is suggested to be increased by mitogen-activated protein kinase (MAPK) pathways, which may be involved in the development of endometriosis [57]. No study, however, has demonstrated a direct link between pro-inflammatory cytokines, endometriosis, and migraines.

Endometriosis is a frequent chronic condition, with a significant impact on women’s quality of life, yet a highly enigmatic disease with unresolved questions in terms of its etiology, pathogenesis, clinical presentation, classifications, diagnosis, and management, or prognosis. Unsupervised learning techniques, like cluster analyses [58], are gaining acceptance in characterizing heterogeneous complex conditions, such as endometriosis [59]. Clustering aggregates similar units (cases or variables) into similar groups, thus partitioning dissimilar objects at a progressively finer granularity. Findings from this novel classification of endometriosis reflect that clusters not only differ with respect to the specific comorbidities used to form them (by design), but also in other relevant clinical characteristics, like age, visits to GP, or referrals to specialists. Clusters of women sharing similar comorbidity profiles may represent diverse “endometriosis trajectories” linked to different plausible but presently unconfirmed pathogenic pathways [60]. An awareness of those comorbidities may facilitate the accurate earlier diagnosis and initiation of treatments targeted toward subgroups of patients. 

This new grouping of comorbidities has identified that endometriosis is associated with health conditions with diverse pathogenic pathways. It is interesting to note that a possible association with endometriosis has already been suggested for all of them. Exploring such associations may provide a deeper insight into the pathophysiology of endometriosis from a multidimensional point of view that should be further investigated from genetic, immunological, and clinical perspectives [61].

The strengths of this work are the large number of cases analyzed, from over 4000 women, and the possibility to analyze population-based data, not restricted to selected populations identified in specialized centers, and generate data-driven endometriosis patient subtypes. 

Our study has several limitations. Data for these analyses were obtained from women with a registered diagnosis of endometriosis in their primary healthcare medical records. Therefore, there may be misclassification bias in the case where women in this database are not definitively diagnosed laparoscopically with a biopsy, which is the gold standard. A possibility to overcome this limitation is the linkage of primary care with hospital data [62]. There may also be an under-diagnosis of cases of women whose GP may not have correctly identified a diagnosis of endometriosis in their medical records. 

The available data do not include the extent of the disease, time since diagnosis, type or degree of symptoms, or treatments received. The associations identified between diseases with endometriosis cannot determine the existence of a temporal association between them: whether endometriosis was diagnosed before or after any of them, or whether they were able to have a longer follow-up of these patients. It was not possible to determine how endometriosis treatment may interfere with these comorbidities or vice versa, nor how the clinical management of comorbidities may affect endometriosis. The diagnostic criteria for all the disorders considered are based on the registration of the physicians in the medical records of the corresponding ICPC-2 codes. A diagnostic bias may also exist for all reported diseases. The lack of information on the severity of the conditions or symptoms prevents the evaluation of possible biological gradients. Furthermore, the prevalence of the disorders discussed here has not been compared to their frequency in the general female population in the age range included in this study. Given the high frequency of several of the comorbidities, it would have been interesting to know the prevalence estimates in women without endometriosis.

This analysis has separated the cases into six clusters, but a different number of clusters may have rendered different results, as well as using another clustering method. This is a common problem in cluster analysis, as no statistical test could indicate the appropriate number of clusters. In this case, we based our decision on the visual assessment of the dendrogram, as well as on the clinical validity of our selection by comparing the distribution of comorbidities if another number of clusters was selected.

The diagnoses are based on ICPC-2, while other electronic record systems code diagnoses using ICD10 codes, which are not perfectly equivalent and, if used, may return different aggregations. The data are limited to Spain and, thus, may not be generalized to other countries. 

In this study, we have included women aged 21–50 years old. Endometriosis is more frequent at younger ages, although it does not disappear after the onset of menopause [59]. However, symptoms of endometriosis are more relevant from 25 to 29 years of age, reaching a maximum peak in the 40–44-year-old group, with a steady decline in older age. Age may influence hormonal activity and be associated with variations in progression or symptoms or diagnostic delays [63].

## 5. Conclusions

Clustering aggregates similar units into similar groups, thus dividing dissimilar objects into other groups with progressively finer granularity—in our case, groups of women with similar comorbidity profiles. There is no single theory that explains all of the heterogeneous and diverse clinical presentations and pathologic features of endometriosis; it is possible that different subtypes develop through different mechanisms and invoke different or partially overlapping theories. Clusters may provide a deeper insight into the multidimensionality of endometriosis, a complex and heterogeneous condition, and may represent various “endometriosis pathways” associated with various plausible but currently unconfirmed pathways of pathogenesis. Mapping the connections between endometriosis and comorbidities may help improve our understanding of the biochemical and molecular mechanisms involved in this disease and outline a more personalized approach to understanding, diagnosing, and treating endometriosis.

## Figures and Tables

**Figure 1 biomedicines-11-02448-f001:**
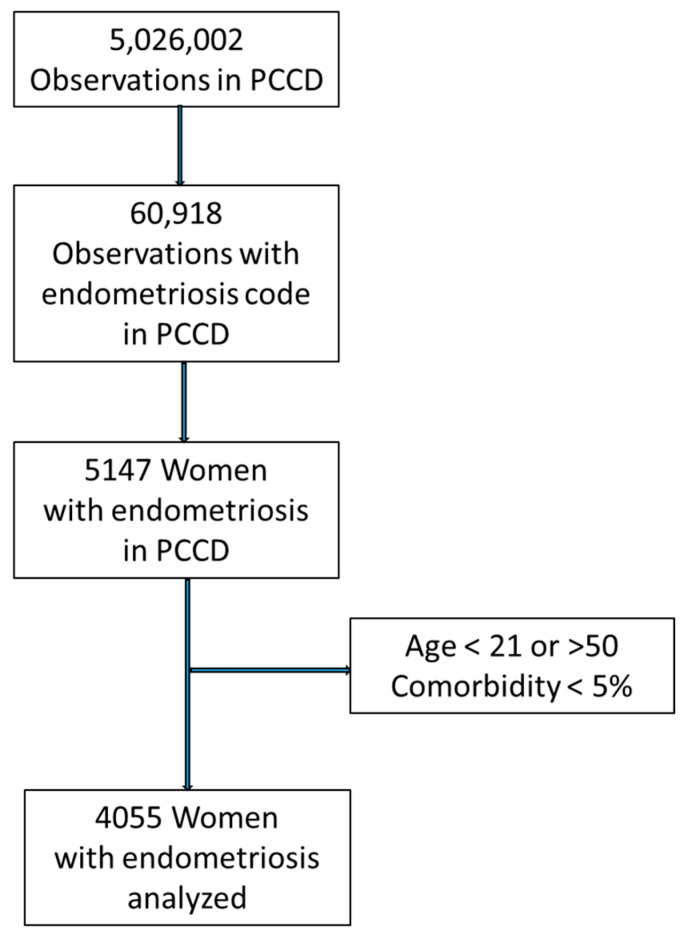
Flowchart.

**Figure 2 biomedicines-11-02448-f002:**
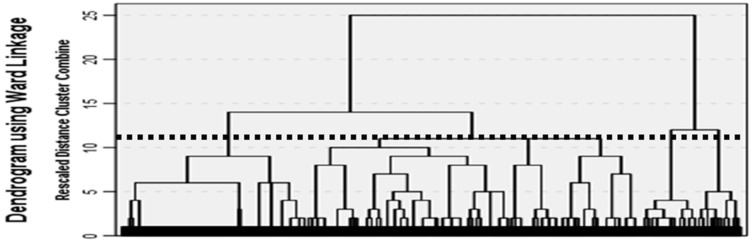
Main characteristics of the clusters of women with endometriosis based on hierarchical clusters. Dotted line represents cut-off values at approximately 11 returning 6 clusters.

**Figure 3 biomedicines-11-02448-f003:**
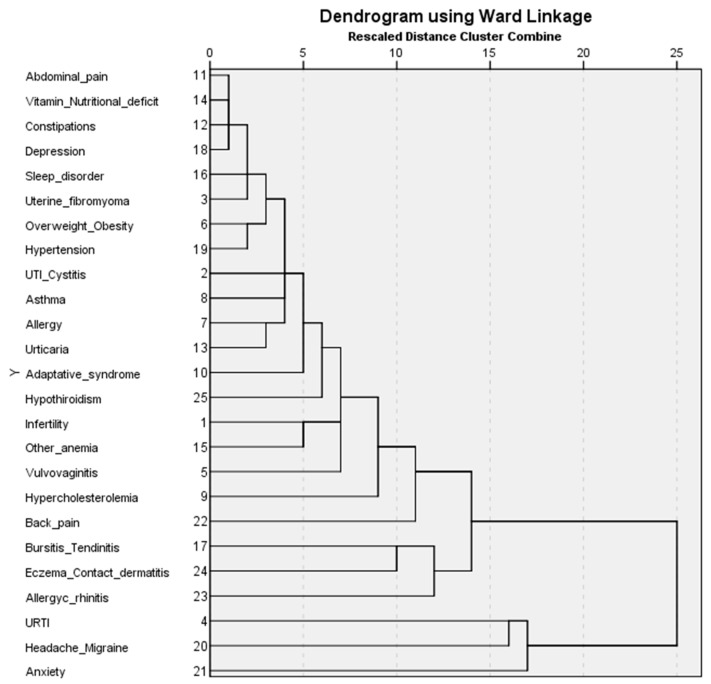
Dendrogram representing clusters of comorbidities of women with endometriosis. UTI: Urinary tract infections, URTI: Upper respiratory tract infections.

**Table 1 biomedicines-11-02448-t001:** Main characteristics of the sample analyzed.

	Cases	%
Age		
Mean		
21–30	368	9.08
31–40	1555	38.35
41–50	2132	52.58
Comorbidities	Cases	%
Infertility	404	9.96
UTI/Cystitis	313	7.72
Uterine fibromyoma	275	6.78
Upper respiratory tract infection	836	20.62
Vulvovaginitis	449	11.07
Obesity	284	7.00
Allergy	321	7.92
Asthma	335	8.26
Chronic/Allergic rhinitis	695	17.14
Thyroid disorders	412	10.16
High cholesterol	527	13.00
Smoking	434	10.70
Adaptation syndrome	370	9.12
Abdominal pain	207	5.10
Constipation	244	6.02
Urticaria	353	8.71
Contact dermatitis/Eczema	625	15.41
Vitamin/Nutrient deficiency	203	5.01
Anemia	415	10.23
Sleep disorder	285	7.03
Back pain	595	14.67
Bursitis/Tendinitis	632	15.59
Depression	255	6.29
Hypertension	298	7.35
Headache/Migraine	1023	25.23
Anxiety	1060	26.14

**Table 2 biomedicines-11-02448-t002:** Main characteristics of the clusters of women with endometriosis based on hierarchical clusters.

	Cluster 1	Cluster 2	Cluster 3	Cluster 4	Cluster 5	Cluster 6
Number of cases	1212	283	1334	534	331	361
%	29.89	6.98	32.90	13.17	8.16	8.90
Visits to general practitioners (mean)	8.1	28.3	13.1	14.1	14.4	14.6
Visits to specialists (mean)	10.2	20.2	11.8	12.4	8.5	14.0
Comorbidities (mean)	1.40	9.72	3.96	4.14	3.82	4.06
Age (mean)	39.9	41.6	41.1	39.5	39.1	39
Age groups (%)						
21–30	10.07	6.36	6.30	12.36	7.25	14.96
31–40	38.94	32.16	34.63	42.13	49.55	39.06
41–50	50.99	61.48	59.07	45.51	43.20	45.98
Comorbidity (%)						
Infertility	2.72	11.66	5.70	7.87	58.91	6.93
Urinary tract infection/Cystitis	13.37	20.85	3.67	4.31	4.53	1.39
Uterine fibromyoma	7.92	15.90	5.85	4.68	7.55	1.66
Upper respiratory tract infection	16.50	51.59	18.44	16.85	21.15	23.27
Vulvovaginitis	3.71	22.97	13.79	9.93	10.57	18.56
Obesity	0.66	26.15	11.62	3.56	5.14	3.05
Allergy	0.99	24.73	4.12	27.34	5.74	5.26
Asthma	0.99	14.84	5.47	35.58	3.63	1.66
Thyroid disorders	3.63	6.71	18.89	6.93	11.78	5.82
High cholesterol	3.71	31.80	17.47	14.04	5.74	18.01
Smoking	3.22	20.85	17.39	8.61	6.34	10.25
Adaptation syndrome	10.15	16.25	7.57	9.55	4.83	9.14
Abdominal pain	5.86	10.25	3.37	5.81	1.81	6.93
Constipation	1.49	18.37	9.30	5.24	2.72	3.60
Urticaria	3.05	24.38	4.12	30.71	3.32	4.71
Vitamin/Nutrient deficiency	4.54	9.54	4.65	6.18	3.93	3.60
Anemia	2.15	31.10	4.65	7.30	54.68	5.26
Sleep disorder	1.49	32.51	9.97	3.93	3.93	2.22
Bursitis/Tendinitis	1.90	44.88	22.26	15.54	9.67	19.39
Depression	0.91	26.86	8.17	4.12	6.04	4.71
Hypertension	1.90	26.15	12.52	2.62	3.63	2.22
Headache/Migraine	7.26	68.55	14.92	23.22	21.15	96.40
Anxiety	4.70	73.85	37.48	23.97	20.24	27.42
Back pain	2.97	43.82	22.56	12.55	13.29	6.37
Chronic/Allergic rhinitis	2.72	48.06	10.42	49.06	16.62	19.39
Contact dermatitis/Eczema	11.39	34.28	12.44	13.30	19.03	24.93

## Data Availability

The data are available from the authors upon request.

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
