# Peer review of "A Novel Classification of Endometriosis Based on Clusters of Comorbidities"

_biomedicines, 2023, doi:10.3390/biomedicines11092448_

Round 1

Reviewer 1 Report

We congratulate the authors for their interest in publishing a new classification of endometriosis. We would like to advance the following points:

1-Line 63: infectious origin (ie, Fusobacterium can be added), cf japanese new research on the topic
2- Line 68 / 69 please make it clear that current classifications do not reflect well pain and infertility. ie a stage 4 patient can be fertile and asymptomatic
3- Line 111 please describe more the STROBE statement.
4- Line 248 what is the definition of subfertility in this population? is it considered a comorbidity or a symptom of endometriosis?
5- Line 280, please rephrase
6- Lin2 319 what do the author propose as future research to deal with this important bias?
7- Line 359. It is better not to include references in the conclusion.

Minor editing of English language required

Author Response

1-Line 63: infectious origin (ie, Fusobacterium can be added), cf japanese new research on the topic

INCLUDED THIS REFERENCE

2- Line 68 / 69 please make it clear that current classifications do not reflect well pain and infertility. ie a stage 4 patient can be fertile and asymptomatic

INCLUDED THIS COMMENT

3- Line 111 please describe more the STROBE statement.

DESCRIBED

4- Line 248 what is the definition of subfertility in this population? is it considered a comorbidity or a symptom of endometriosis?

SUBFERTILITY WAS NOT CORRECTLY TRANSLATED FROM ICPC-2.

AN EXPLANATION OF THE CONCEPT OF COMORBDITIES FOR THE PURPOSE OF THIS WORK IN NOW IN THE METHODS SECTION

5- Line 280, please rephrase

DONE

6- Lin2 319 what do the author propose as future research to deal with this important bias?

INCLUDED A COMMENT

7- Line 359. It is better not to include references in the conclusion.

CORRECTED

Reviewer 2 Report

The authors classified endometriosis with comorbidities using the (hierarchical) cluster analysis method. In 4055 women with endometriosis, 84% of women had at least 1 comorbidity. They classified these women into six clusters according to their comorbidities. Comorbidities-based clusters may be important to scientific study of endometriosis, contributing to clarifying its clinical complexity and variability.

This article is informative and useful for future clinical and pathological studies of endometriosis.

It might be improved on the following issues.

1.    Introduction: line 94: “not only ovarian or endometrial cancers”. This part should be “not only ovarian but also endometrial cancers.”

2.    And are there any case of cancer arising from endometriosis?

3.    Result: line 154: “analysis of 27 comorbidities”: In Table 1, 2 and Figure 3, there are only 26 symptoms are shown. I would like the authors to clarify this pint.

Minor editing of English language required.

Author Response

  1. Introduction: line 94: “not only ovarian or endometrial cancers”. This part should be “not only ovarian but also endometrial cancers.”

CORRECTED

  1. And are there any case of cancer arising from endometriosis?

THERE IS SOME LITERATURE SHOWING THERE MAY BE AN ASSOCIATION BETWEEN ENDOMETRIOSIS AND CERTAIN CANCER TYPES, BUT EXPLORING THOSE CONNECTIONS IS NOT POSSIBLE WITH THIS DATA.

  1. Result: line 154: “analysis of 27 comorbidities”: In Table 1, 2 and Figure 3, there are only 26 symptoms are shown. I would like the authors to clarify this pint.

CORRECTED, THEY ARE 26. THE DEFINITION OF COMORBIDITIES IS INCLUDED IN THE METHODS SECTION.
